# Optimal transport maps for distribution preserving operations on latent spaces of Generative Models

**Eirikur Agustsson, Alexander Sage, Radu Timofte & Luc Van Gool**
Computer Vision Lab
ETH Zurich
Switzerland
{aeirikur,sagea,timofter,vangool}@vision.ee.ethz.ch

## Abstract

Generative models such as Variational Auto Encoders (VAEs) and Generative Adversarial Networks (GANs) are typically trained for a fixed prior distribution in the latent space, such as uniform or Gaussian. After a trained model is obtained, one can sample the Generator in various forms for exploration and understanding, such as interpolating between two samples, sampling in the vicinity of a sample or exploring differences between a pair of samples applied to a third sample. However, the latent space operations commonly used in the literature so far induce a distribution mismatch between the resulting outputs and the prior distribution the model was trained on. Previous works have attempted to reduce this mismatch with heuristic modification to the operations or by changing the latent distribution and re-training models. In this paper, we propose a framework for modifying the latent space operations such that the distribution mismatch is fully eliminated. Our approach is based on optimal transport maps, which adapt the latent space operations such that they fully match the prior distribution, while minimally modifying the original operation. Our matched operations are readily obtained for the commonly used operations and distributions and require no adjustment to the training procedure.

## 1 Introduction & Related Work

Generative models such as Variational Autoencoders (VAEs) (Kingma & Welling, 2013) and Generative Adversarial Networks (GANs) (Goodfellow et al., 2014) have emerged as popular techniques for unsupervised learning of intractable distributions. In the framework of Generative Adversarial Networks (GANs) (Goodfellow et al., 2014), the generative model is obtained by jointly training a generator $G$ and a discriminator $D$ in an adversarial manner. The discriminator is trained to classify synthetic samples from real ones, whereas the generator is trained to map samples drawn from a fixed prior distribution to synthetic examples which fool the discriminator. Variational Autoencoders (VAEs) (Kingma & Welling, 2013) are also trained for a fixed prior distribution, but this is done through the loss of an Autoencoder that minimizes the variational lower bound of the data likelihood. For both VAEs and GANs, using some data $\mathcal{X}$ we end up with a trained generator $G$, that is supposed to map latent samples $\boldsymbol{z}$ from the fixed prior distribution to output samples $G(\boldsymbol{z})$ which (hopefully) have the same distribution as the data.

In order to understand and visualize the learned model $G(z)$, it is a common practice in the literature of generative models to explore how the output $G(z)$ behaves under various arithmetic operations on the latent samples $z$. However, the operations typically used so far, such as linear interpolation (Goodfellow et al., 2014), spherical interpolation (White, 2016), vicinity sampling and vector arithmetic (Radford et al., 2015), cause a distribution mismatch between the latent prior distribution and the results of the operations. This is problematic, since the generator $G$ was trained on a fixed prior and expects to see inputs with statistics consistent with that distribution.

To address this, we propose to use distribution matching transport maps, to obtain analogous latent space operations (e.g. interpolation, vicinity sampling) which preserve the prior distribution of

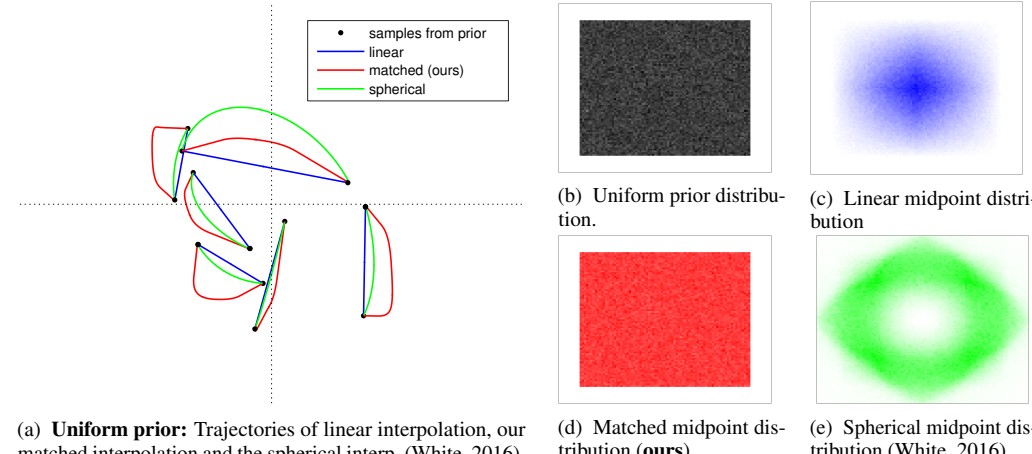

(a) **Uniform prior:** Trajectories of linear interpolation, our matched interpolation and the spherical interp. (White, 2016).

(b) Uniform prior distribution.

(c) Linear midpoint distribution

(d) Matched midpoint distribution (**ours**)

(e) Spherical midpoint distribution (White, 2016)

Figure 1: We show examples of distribution mismatches induced by the previous interpolation schemes when using a uniform prior in two dimensions. Our matched interpolation avoids this with a minimal modification to the linear trajectory, traversing through the space such that all points along the path are distributed identically to the prior.

the latent space, while minimally changing the original operation. In Figure 1 we showcase how our proposed technique gives an interpolation operator which avoids distribution mismatch when interpolating between samples of a uniform distribution. The points of the (red) matched trajectories are obtained as minimal deviations (in expectation of $l_1$ distance) from the the points of the (blue) linear trajectory.

## 1.1 Distribution Mismatch and Related Approaches

In the literature there are dozens of papers that use sample operations to explore the learned models (Bengio et al. (2013); Goodfellow et al. (2014); Dosovitskiy et al. (2015); Reed et al. (2016); Brock et al. (2016); Reed et al. (2016) to name a few), but most of them have ignored the problem of distribution mismatch. Kingma & Welling (2013) and Makhzani et al. (2015) sidestep the problem when visualizing their models, by not performing operations on latent samples, but instead restrict the latent space to 2-d and uniformly sample the percentiles of the distribution on a 2-d grid. This way, the samples have statistics that are consistent with the prior distribution. However, this approach does not scale up to higher dimensions - whereas the latent spaces used in the literature can have hundreds of dimensions.

White (2016) experimentally observe that there is a distribution mismatch between the norm for points drawn from uniform or Gaussian distribution and points obtained with linear interpolation (SLERP), and (heuristically) propose to use a so-called *spherical linear interpolation* to reduce the mismatch, obtaining higher quality interpolated samples.

While SLERP has been subjectively observed to produce better looking samples than linear interpolation and is now commonly, its heuristic nature has limited it from fully replacing the linear interpolation. Furthermore, while perhaps possible it is not obvious how to generalize it to other operations, such as vicinity sampling, n-point interpolation and random walk. In Section 2 we show that for interpolation, in high dimensions SLERP tends to approximately perform distribution matching the approach taken by our framework which can explain why it works well in practice.

Kilcher et al. (2018) further analyze the (norm) distribution mismatch observed by White (2016) (in terms of KL-Divergence) for the special case of Gaussian priors, and propose an alternative prior distribution with dependent components which produces less (but still nonzero) distribution mismatch for linear interpolation, at the cost of needing to re-train and re-tune the generative models.

In contrast, we propose a framework which allows one to adapt generic operations, such that they fully preserve the original prior distribution while being faithful to the original operation. Thus the KL-Divergence between the prior and the distribution of the results from our operations is **zero**.

The approach works as follows: we are given a 'desired' operation, such as linear interpolation $\boldsymbol{y} = t\boldsymbol{z}_1 + (1-t)\boldsymbol{z}_2, t \in [0,1]$. Since the distribution of $\boldsymbol{y}$ does not match the prior distribution of $\boldsymbol{z}$, we search for a warping $f : \mathbb{R}^d \to \mathbb{R}^d$, such that $\tilde{\boldsymbol{y}} = f(\boldsymbol{y})$ has the same distribution as $\boldsymbol{z}$. In order to have the modification $\tilde{\boldsymbol{y}}$ as faithful as possible to the original operation $\boldsymbol{y}$, we use optimal transform

| Operation | Expression |
|-----------|------------|
| *2-point interpolation* | $\boldsymbol{y} = t\boldsymbol{z}_1 + (1 - t)\boldsymbol{z}_2$ , $t \in [0, 1]$ |
| *n-point interpolation* | $\boldsymbol{y} = \sum_{i=1}^{n} t_i \boldsymbol{z}_i$ with $\sum_i t_i = 1$ |
| *Vicinity sampling* | $\boldsymbol{y}_j = \boldsymbol{z}_1 + \epsilon \boldsymbol{u}_j$ for $j = 1, \cdots, k$ |
| *Analogies* | $\boldsymbol{y} = \boldsymbol{z}_3 + (\boldsymbol{z}_2 - \boldsymbol{z}_1)$ |

Table 1: Examples of interesting sample operations which need to be adapted ('matched') if we want the distribution of the result $\boldsymbol{y}$ to match the prior distribution.

maps (Santambrogio, 2015; Villani, 2003; 2008) to find a minimal modification of $\boldsymbol{y}$ which recovers the prior distribution $\boldsymbol{z}$.

This is illustrated in Figure 1a, where each point $\tilde{\boldsymbol{y}}$ of the matched curve is obtained by warping a corresponding point $\boldsymbol{y}$ of the linear trajectory, while not deviating too far from the line.

## 2 FROM DISTRIBUTION MISMATCH TO OPTIMAL TRANSPORT

With implicit models such as GANs (Goodfellow et al., 2014) and VAEs (Kingma & Welling, 2013), we use the data $\mathcal{X}$, drawn from an unknown random variable $\boldsymbol{x}$, to learn a generator $G : \mathbb{R}^d \mapsto \mathbb{R}^{d'}$ with respect to a fixed prior distribution $p_{\boldsymbol{z}}$, such that $G(\boldsymbol{z})$ approximates $\boldsymbol{x}$. Once the model is trained, we can sample from it by feeding latent samples $\boldsymbol{z}$ through $G$.

We now bring our attention to *operations* on latent samples $\boldsymbol{z}_1, \cdots, \boldsymbol{z}_k$ from $p_{\boldsymbol{z}}$, i.e. mappings

$$\kappa : \mathbb{R}^d \times \cdots \times \mathbb{R}^d \to \mathbb{R}^d. \tag{1}$$

We give a few examples of such operations in Table 1.

Since the inputs to the operations are random variables, their output $\boldsymbol{y} = \kappa(\boldsymbol{z}_1, \cdots, \boldsymbol{z}_k)$ is also a random variable (commonly referred to as a *statistic*). While we typically perform these operations on *realized* (i.e. observed) samples, our analysis is done through the underlying random variable $\boldsymbol{y}$. The same treatment is typically used to analyze other statistics over random variables, such as the sample mean, sample variance and test statistics.

In Table 1 we show example operations which have been commonly used in the literature. As discussed in the Introduction, such operations can provide valuable insight into how the trained generator $G$ changes as one creates related samples $\boldsymbol{y}$ from some source samples. The most common such operation is the linear interpolation, which we can view as an operation

$$\boldsymbol{y}_t = t\boldsymbol{z}_1 + (1 - t)\boldsymbol{z}_2, \tag{2}$$

where $\boldsymbol{z}_1, \boldsymbol{z}_2$ are latent samples from the prior $p_{\boldsymbol{z}}$ and $\boldsymbol{y}_t$ is parameterized by $t \in [0, 1]$.

Now, assume $\boldsymbol{z}_1$ and $\boldsymbol{z}_2$ are i.i.d, and let $Z_1, Z_2$ be their (scalar) first components with distribution $p_Z$. Then the first component of $\boldsymbol{y}_t$ is $Y_t = tZ_1 + (1 - t)Z_2$, and we can compute:

$$\text{Var}[Y_t] = \text{Var}[tZ_1 + (1 - t)Z_2] = t^2 \text{Var}[Z_1] + (1 - t)^2 \text{Var}[Z_2] = (1 + 2t(t - 1))\text{Var}[Z]. \tag{3}$$

Since $(1 + 2t(t - 1)) \neq 1$ for all $t \in [0, 1] \setminus \{0, 1\}$, it is in general impossible for $\boldsymbol{y}_t$ to have the same distribution as $\boldsymbol{z}$, which means that distribution mismatch is *inevitable* when using linear interpolation. A similar analysis reveals the same for all of the operations in Table 1.

This leaves us with a dilemma: we have various intuitive operations (see Table 1) which we would want to be able to perform on samples, but their resulting distribution $p_{\boldsymbol{y}_t}$ is inconsistent with the distribution $p_{\boldsymbol{z}}$ we trained $G$ for.

Due to the *curse of dimensionality*, as empirically observed by White (2016), this mismatch can be significant in high dimensions. We illustrate this in Figure 2, where we plot the distribution of the squared norm $\|\boldsymbol{y}_t\|^2$ for the midpoint $t = 1/2$ of linear interpolation, compared to the prior distribution $\|\boldsymbol{z}\|^2$. With $d = 100$ (a typical dimensionality for the latent space), the distributions are dramatically different, having almost no common support. Kilcher et al. (2018) quantify this mismatch for Gaussian priors in terms of KL-Divergence, and show that it grows linearly with the dimension $d$. In Appendix A (see Supplement) we expand this analysis and show that this happens for all prior distributions with i.i.d. entries (i.e. not only Gaussian), both in terms of geometry and KL-Divergence.

### 2.1 DISTRIBUTION MATCHING WITH OPTIMAL TRANSPORT

In order to address the distribution mismatch, we propose a simple and intuitive framework for constructing distribution preserving operators, via optimal transport:

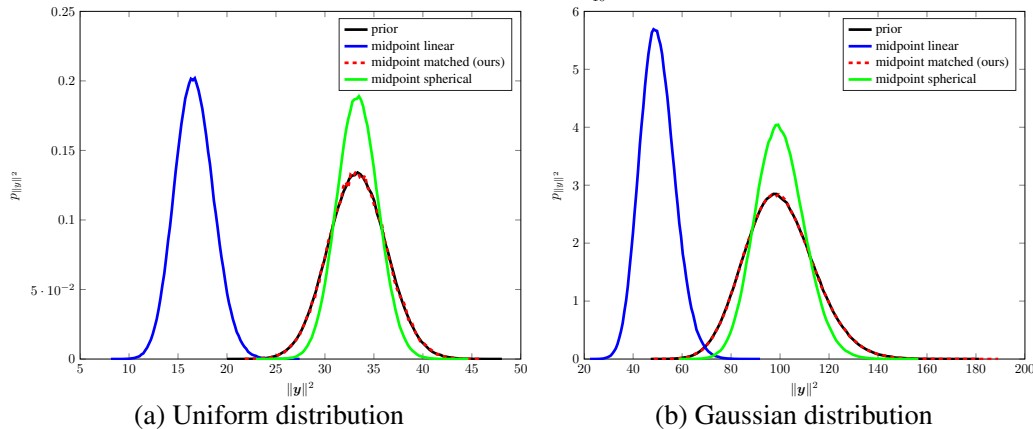

| (a) Uniform distribution | (b) Gaussian distribution |

Figure 2: Distribution of the squared norm $\|\boldsymbol{y}\|^2$ of midpoints for two prior distributions in 100 dimensions: (a) components uniform on $[-1, 1]$ and (b) components Gaussian $\mathcal{N}(0, 1)$, for linear interpolation, our proposed matched interpolation and the spherical interpolation proposed by White (2016). Both linear and spherical interpolation introduce a distribution mismatch, whereas our proposed matched interpolation preserves the prior distribution for both priors.

**Strategy 1** (Optimal Transport Matched Operations).

1. *We construct an 'intuitive' operator $\boldsymbol{y} = \kappa(\boldsymbol{z}_1, \cdots, \boldsymbol{z}_k)$.*

2. *We analytically (or numerically) compute the resulting (mismatched) distribution $p_{\boldsymbol{y}}$*

3. *We search for a minimal modification $\tilde{\boldsymbol{y}} = f(\boldsymbol{y})$ (in the sense that $E_{\boldsymbol{y}}[c(\tilde{\boldsymbol{y}}, \boldsymbol{y})]$ is minimal with respect to a cost c), such that distribution is brought back to the prior, i.e. $p_{\tilde{\boldsymbol{y}}} = p_{\boldsymbol{z}}$.*

The cost function in step 3 could e.g. be the euclidean distance $c(x, y) = \|x - y\|$, and is used to measure how faithful the modified operator, $\tilde{\boldsymbol{y}} = f(\kappa(\boldsymbol{z}_1, \cdots, \boldsymbol{z}_k))$ is to the original operator $k$. Finding the map $f$ which gives a minimal modification can be challenging, but fortunately it is a well studied problem from optimal transport theory. We refer to the modified operation $\tilde{\boldsymbol{y}}$ as the *matched* version of $\boldsymbol{y}$, with respect to the cost $c$ and prior distribution $p_{\boldsymbol{z}}$.

For completeness, we introduce the key concepts of optimal transport theory in a simplified setting, i.e. assuming probability distributions are in euclidean space and skipping measure theoretical formalism. We refer to Villani (2003; 2008) and Santambrogio (2015) for a thorough and formal treatment of optimal transport.

The problem of step (3) above was first posed by Monge (1781) and can more formally be stated as:

**Problem 1** (Santambrogio (2015) Problem 1.1). *Given probability distributions $p_{\boldsymbol{x}}, p_{\boldsymbol{y}}$, with domains $\mathcal{X}, \mathcal{Y}$ respectively, and a cost function $c : \mathcal{X} \times \mathcal{Y} \to \mathbb{R}^+$, we want to minimize*

$$inf \left\{ E_{\boldsymbol{x} \sim p_{\boldsymbol{x}}}[c(\boldsymbol{x}, f(\boldsymbol{x}))] \Big| f : \mathcal{X} \to \mathcal{Y}, f(\boldsymbol{x}) \sim p_{\boldsymbol{y}} \right\} \tag{MP}$$

*We refer to the minimizer $f^* \mathcal{X} \to \mathcal{Y}$ of* (MP) *(if it exists), as the optimal transport map from $p_{\boldsymbol{x}}$ to $p_{\boldsymbol{y}}$ with respect to the cost $c$.*

However, the problem remained unsolved until a relaxed problem was studied by Kantorovich (1942):

**Problem 2** (Santambrogio (2015) Problem 1.2). *Given probability distributions $p_{\boldsymbol{x}}, p_{\boldsymbol{y}}$, with domains $\mathcal{X}, \mathcal{Y}$ respectively, and a cost function $c : \mathcal{X} \times \mathcal{Y} \to \mathbb{R}^+$, we want to minimize*

$$inf \left\{ E_{(\boldsymbol{x}, \boldsymbol{y}) \sim p_{\boldsymbol{x}, \boldsymbol{y}}}[c(\boldsymbol{x}, \boldsymbol{y})] \Big| (\boldsymbol{x}, \boldsymbol{y}) \sim p_{\boldsymbol{x}, \boldsymbol{y}}, \boldsymbol{x} \sim p_{\boldsymbol{x}}, \boldsymbol{y} \sim p_{\boldsymbol{y}} \right\}, \tag{KP}$$

*where $(\boldsymbol{x}, \boldsymbol{y}) \sim p_{\boldsymbol{x}, \boldsymbol{y}}, \boldsymbol{x} \sim p_{\boldsymbol{x}}, \boldsymbol{y} \sim p_{\boldsymbol{y}}$ denotes that $(\boldsymbol{x}, \boldsymbol{y})$ have a joint distribution $p_{\boldsymbol{x}, \boldsymbol{y}}$ which has (previously specified) marginals $p_{\boldsymbol{x}}$ and $p_{\boldsymbol{y}}$.*

*We refer to the joint $p_{\boldsymbol{x}, \boldsymbol{y}}$ which minimizes* (KP) *as the optimal transport plan from $p_{\boldsymbol{x}}$ to $p_{\boldsymbol{y}}$ with respect to the cost $c$.*

The key difference is to relax the deterministic relationship between $\boldsymbol{x}$ and $f(\boldsymbol{x})$ to a joint probability distribution $p_{\boldsymbol{x}, \boldsymbol{y}}$ with marginals $p_{\boldsymbol{x}}$ and $p_{\boldsymbol{y}}$ for $\boldsymbol{x}$ and $\boldsymbol{y}$. In the case of Problem 1, the minimization

might be over the empty set since it is not guaranteed that there exists a mapping $f$ such that $f(\boldsymbol{x}) \sim \boldsymbol{y}$. In contrast, for Problem 2, one can always construct a joint density $p_{\boldsymbol{x},\boldsymbol{y}}$ with $p_{\boldsymbol{x}}$ and $p_{\boldsymbol{y}}$ as marginals, such as the trivial construction where $\boldsymbol{x}$ and $\boldsymbol{y}$ are independent, i.e. $p_{\boldsymbol{x},\boldsymbol{y}}(x,y) := p_{\boldsymbol{x}}(x)p_{\boldsymbol{y}}(y)$.

Note that given a joint density $p_{\boldsymbol{x},\boldsymbol{y}}(x,y)$ over $\mathcal{X} \times \mathcal{Y}$, we can view $\boldsymbol{y}$ conditioned on $\boldsymbol{x} = x$ for a fixed $x$ as a stochastic function $\boldsymbol{f}(x)$ from $\mathcal{X}$ to $\mathcal{Y}$, since given a fixed $x$ do not get a specific function value $f(x)$ but instead a random variable $\boldsymbol{f}(x)$ that depends on $x$, with $\boldsymbol{f}(x) \sim \boldsymbol{y}|\boldsymbol{x} = x$ with density $p_{\boldsymbol{y}}(y|\boldsymbol{x} = x) := \frac{p_{\boldsymbol{x},\boldsymbol{y}}(x,y)}{p_{\boldsymbol{x}}(x)}$. In this case we have $(\boldsymbol{x}, \boldsymbol{f}(\boldsymbol{x})) \sim p_{\boldsymbol{x},\boldsymbol{y}}$, so we can view the Problem KP as a relaxation of Problem MP where $f$ is allowed to be a stochastic mapping.

While the relaxed problem of Kantorovich (KP) is much more studied in the optimal transport literature, for our purposes of constructing operators it is desirable for the mapping $f$ to be deterministic as in (MP) (see Appendix C for a more detailed discussion on deterministic vs stochastic operations).

To this end, we will choose the cost function $c$ such that the two problems coincide and where we can find an analytical solution $f$ or at least an efficient numerical solution.

In particular, we note that the operators in Table 1 are all *pointwise*, such that if the points $\boldsymbol{z}_i$ have i.i.d. components, then the result $\boldsymbol{y}$ will also have i.i.d. components.

If we combine this with the constraint for the cost $c$ to be additive over the components of $\boldsymbol{x}, \boldsymbol{y}$, we obtain the following simplification:

**Theorem 1.** *Suppose $p_{\boldsymbol{x}}$ and $p_{\boldsymbol{y}}$ have i.i.d components and $c$ over $\mathcal{X} \times \mathcal{Y} = \mathbb{R}^d \times \mathbb{R}^d$ decomposes as*

$$c(x,y) = \sum_{i=1}^{d} C(x^{(i)}, y^{(i)}). \tag{4}$$

*Consequently, the minimization problems (MP) and (KP) turn into $d$ identical scalar problems for the distributions $p_X$ and $p_Y$ of the components of $\boldsymbol{x}$ and $\boldsymbol{y}$:*

$$inf\left\{ E_{X \sim p_X}[C(X, T(X))] \Big| T : \mathbb{R} \to \mathbb{R}, T(X) \sim p_Y \right\} \tag{MP-1-D}$$

$$inf\left\{ E_{(X,Y) \sim p_{X,Y}}[C(X,Y)] \Big| (X,Y) \sim p_{X,Y}, X \sim p_X, Y \sim p_Y \right\}, \tag{KP-1-D}$$

*such that an optimal transport map $T$ for (MP-1-D) gives an optimal transport map $f$ for (MP) by pointwise application of $T$, i.e. $f(x)^{(i)} := T(x^{(i)})$, and an optimal transport plan $p_{X,Y}$ for (KP-1-D) gives an optimal transport plan $p_{\boldsymbol{x},\boldsymbol{y}}(x,y) := \prod_{i=1}^{d} p_{X,Y}(x^{(i)}, y^{(i)})$ for (KP).*

*Proof.* See Appendix. $\qquad\qquad\square$

Fortunately, under some mild constraints, the scalar problems have a known solution:

**Theorem 2** (Theorem 2.9 in Santambrogio (2015)). *Let $h : \mathbb{R} \to \mathbb{R}^+$ be convex and suppose the cost $C$ takes the form $C(x,y) = h(x - y)$. Given an continuous source distribution $p_X$ and a target distribution $p_Y$ on $\mathbb{R}$ having a finite optimal transport cost in (KP-1-D), then*

$$T_{X \to Y}^{mon}(x) := F_Y^{[-1]}(F_X(x)), \tag{5}$$

*defines an optimal transport map from $p_X$ to $p_Y$ for (MP-1-D), where $F_X(x) := \int_{-\infty}^{x} p_X(x')dx'$ is the Cumulative Distribution Function (CDF) of $X$ and $F_Y^{[-1]}(y) := \inf\{t \in \mathbb{R}|F_Y(t) \geq y\}$ is the pseudo-inverse of $F_Y$. Furthermore, the joint distribution of $(X, T_{X \to Y}^{mon}(X))$ defines an optimal transport plan for (KP-1-D).*

The mapping $T_{X \to Y}^{\mathrm{mon}}(x)$ in Theorem 2 is non-decreasing and is known as the *monotone transport map* from $X$ to $Y$. It is easy to verify that $T_{X \to Y}^{\mathrm{mon}}(X)$ has the distribution of $Y$, in particular $F_X(X) \sim \mathrm{Uniform}(0,1)$ and if $U \sim \mathrm{Uniform}(0,1)$ then $F_Y^{[-1]}(U) \sim Y$.

Now, combining Theorems 1 and 2, we obtain a concrete realization of the Strategy 1 outlined above. We choose the cost $c$ such that it admits to Theorem 1, such as $c(\boldsymbol{x}, \boldsymbol{y}) := \|\boldsymbol{x} - \boldsymbol{y}\|_1$, and use an operation that is pointwise, so we just need to compute the monotone transport map in (5). That is, if

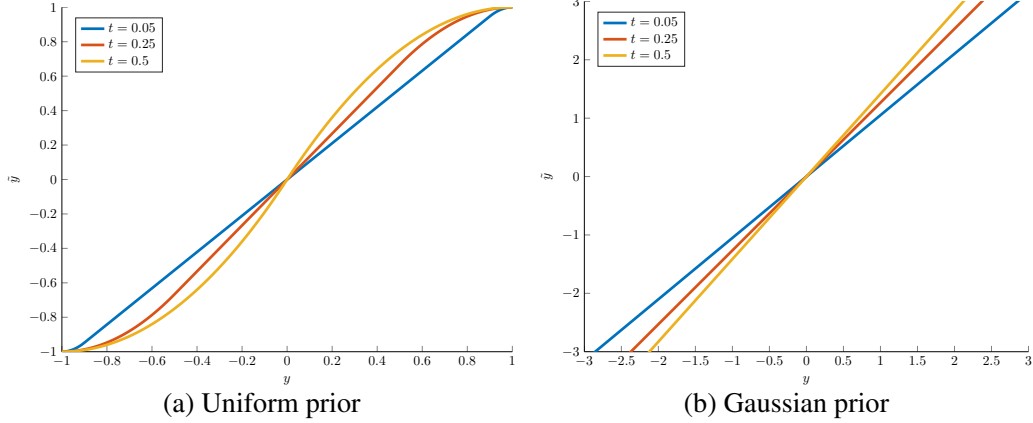

(a) Uniform prior                       (b) Gaussian prior

Figure 3: We show the monotone transport maps for linear interpolation evaluated at $t \in \{0.05, 0.25, 0.5\}$, to Uniform and Gaussian priors.

$z$ has i.i.d components with distribution $p_Z$, we just need to compute the component distribution $p_Y$ of the result $y$ of the operation, the CDFs $F_Z, F_Y$ and obtain

$$T_{Y \to Z}^{\mathrm{mon}}(y) := F_Z^{[-1]}(F_Y(y)) \tag{6}$$

as the component-wise modification of $y$, i.e. $\tilde{y}^{(i)} := T_{Y \to Z}^{\mathrm{mon}}(y^{(i)})$.

In Figure 3 we show the monotone transport map for the linear interpolation $y = tz_1 + (1-t)z_2$ for various values of $t$. The detailed calculations and examples for various operations are given in Appendix B, for both Uniform and Gaussian priors.

## 3 SIMULATIONS

To validate the correctness of the matched operators computed in Appendix B, we numerically simulate the distributions for toy examples, as well as prior distributions typically used in the literature.

**Priors vs. interpolations in** $2$**-D** For Figure 1, we sample 1 million pairs of points in two dimension, from a uniform prior (on $[-1, 1]^2$), and estimate numerically the midpoint distribution of linear interpolation, our proposed matched interpolation and the spherical interpolation of White (2016). It is reassuring to see that the matched interpolation gives midpoints which are identically distributed to the prior. In contrast, the linear interpolation condenses more towards the origin, forming a pyramid-shaped distribution (the result of convolving two boxes in 2-d). Since the spherical interpolation of White (2016) follows a great circle with varying radius between the two points, we see that the resulting distribution has a "hole" in it, "circling" around the origin for both priors.

**Priors vs. interpolations in** $100$**-D** For Figure 2, we sample 1 million pairs of points in $d = 100$ dimensions, using either i.i.d. uniform components on $[-1, 1]$ or Gaussian $\mathcal{N}(0, 1)$ and compute the distribution of the squared norm of the midpoints. We see there is a dramatic difference between vector lengths in the prior and the midpoints of linear interpolation, with only minimal overlap. We also see that the spherical interpolation (SLERP) is approximately matching the prior (norm) distribution, having a matching first moment, but otherwise also induces a distribution mismatch. In contrast, our matched interpolation, fully preserves the prior distribution and perfectly aligns. We note that this setting ($d = 100$, uniform or Gaussian) is commonly used in the literature.

## 4 EXPERIMENTS

**Setup** We used DCGAN (Radford et al., 2015) generative models trained on LSUN bedrooms (Yu et al., 2015), CelebA (Liu et al., 2015) and LLD (Sage et al., 2017; 2018), an icon dataset, to qualitatively evaluate. For LSUN, the model was trained for two different output resolutions, providing $64 \times 64$ pixel and a $128 \times 128$ pixel output images (where the latter is used in figures containing larger sample images). The models for LSUN and the icon dataset where both trained on a uniform latent prior distribution, while for CelebA a Gaussian prior was used. The dimensionality of the latent space is 100 for both LSUN and CelebA, and 512 for the model trained on the icon model. Furthermore we use improved Wasserstein GAN (iWGAN) with gradient penalty (Gulrajani et al.,

| Dataset | CIFAR-10 | LLD-icon | LSUN | CelebA |
|---|---|---|---|---|
| **Model** | iWGAN | DCGAN | DCGAN | DCGAN |
| **Prior** | Gaussian, 128-D | Uniform, 100-D | Uniform, 100-D | Gaussian, 100-D |
| **Inception scores for midpoints:** | | | | |
| random samples | **7.90** ± 0.11 | **3.70** ± 0.09 | **3.90** ± 0.08 | **2.05** ± 0.04 |
| 2-point linear | 7.12 ± 0.08 (-10%) | 3.56 ± 0.06 (-4%) | 3.57 ± 0.07 (-8%) | 1.71 ± 0.02 (-17%) |
| 2-point matched | **7.89** ± 0.08 | **3.69** ± 0.08 | **3.89** ± 0.08 | **2.04** ± 0.03 |
| 4-point linear | 5.84 ± 0.08 (-26%) | 3.45 ± 0.08 (-7%) | 2.95 ± 0.06 (-24%) | 1.46 ± 0.01 (-29%) |
| 4-point matched | **7.91** ± 0.09 | **3.69** ± 0.10 | **3.91** ± 0.10 | **2.04** ± 0.04 |

Table 2: Inception scores on LLD-icon, LSUN, CIFAR-10 and CelebA for the midpoints of linear interpolation and its matched counterpart. Scores are reported as mean ± standard deviation (relative change in %). Our matched variants fully recover from the (up to 29%) score drop of the linear interpolation, giving the same quality as random samples.

| Prior | Perturbation $\|\cdot\|_2$ | Perturbation $\|\cdot\|_1$ |
|---|---|---|
| Gaussian, 100-D | 0.2463 | 0.2460 |
| Uniform, 100-D | 0.2377 | 0.2477 |
| Gaussian, 128-D | 0.2470 | 0.2460 |
| Uniform, 128-D | 0.2384 | 0.2479 |

Table 3: We measure over the average (normalized) perturbation $\|\tilde{\boldsymbol{y}} - \boldsymbol{y}\|_p / \|\boldsymbol{y}\|_p$ incurred by our matched interpolation for the latent spaces used in Table 2, for $p = 1, 2$.

2017) trained on CIFAR-10 at $32 \times 32$ pixels with a 128-dimensional Gaussian prior to compute inception scores.

## 4.1 QUANTITATIVE RESULTS

To measure the effect of the distribution mismatch, we quantitatively evaluate using the Inception score(Salimans et al., 2016). In Table 2 we compare the Inception score of our trained models (i.e. using random samples from the prior) with the score when sampling midpoints from the 2-point and 4-point interpolations described above, reporting mean and standard deviation with 50,000 samples, as well as relative change to the original model scores if they are significant. Compared to the original scores of the trained models (random samples), our matched operations are statistically indistinguishable (as expected) while the linear interpolation gives a significantly lower score in all settings (up to 29% lower).

However, this is not surprising, since our matched operations are *guaranteed* to produce samples that come from the same distribution as the random samples.

To quantify the effect our matching procedure has on the original operation, in Table 3 we compute the perturbation incurred when warping the linear interpolation $\boldsymbol{y}$ to the matched counterpart $\tilde{y}$ for 2-point interpolation on the latent spaces used in Table 2. We compute the normalized perturbation $\|\tilde{\boldsymbol{y}}_t - \boldsymbol{y}_t\|_p / \|\boldsymbol{y}_t\|_p$ (with $p = 1$ corresponding to $l_1$ distance and $p = 2$ to $l_2$ distance), over $N = 100000$ interpolation points $\boldsymbol{y}_t = t\boldsymbol{z}_1 + (1 - t)\boldsymbol{z}_2$ where $\boldsymbol{z}_1, \boldsymbol{z}_2$ are sampled from the prior and $t \in [0, 1]$ sampled uniformly. We observe that for all priors and both metrics, the perturbation is in the range $0.23 - 0.25$, i.e. less than a one fourth of $\|\boldsymbol{y}_t\|$.

## 4.2 QUALITATIVE RESULTS

In the following, we will qualitatively show that our matched operations behave as expected, and that there is a visual difference between the original operations and the matched counterparts. To this end, the generator output for latent samples produced with linear interpolation, SLERP (spherical linear interpolation) of White (2016) and our proposed matched interpolation will be compared.

**2-point interpolation**  We begin with the classic example of 2-point interpolation: Figure 4 shows three examples per dataset for an interpolation between 2 points in latent space. Each example is first done via linear interpolation, then SLERP and finally matched interpolation. It is immediately obvious in Figures 4a and 4b that linear interpolation produces inferior results with generally more blurry, less saturated and less detailed output images.

The SLERP heuristic and matched interpolation are slightly different visually, but we do not observe a difference in visual quality. However, we stress that the goal of this work is to construct operations in a principled manner, whose samples are consistent with the generative model. In the case of linear

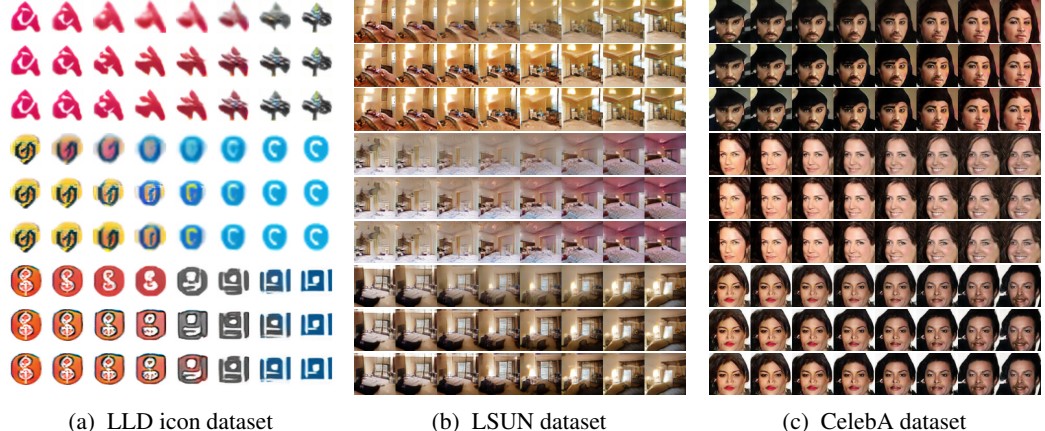

(a) LLD icon dataset       (b) LSUN dataset       (c) CelebA dataset

Figure 4: 2-point interpolation: Each example shows linear, SLERP and transport matched interpolation from top to bottom respectively. For LLD icon dataset (a) and LSUN (b), outputs are produced with DCGAN using a uniform prior distribution, whereas the CelebA model (c) uses a Gaussian prior. The output resolution for the (a) is $32 \times 32$, for (b) and (c) $64 \times 64$ pixels.

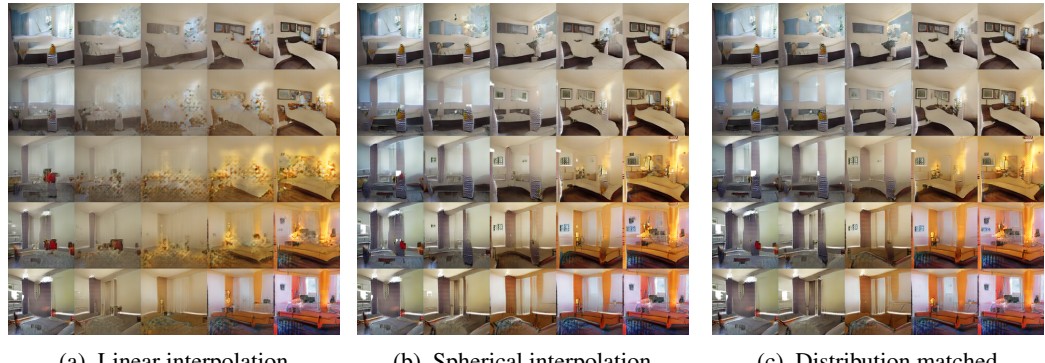

(a) Linear interpolation       (b) Spherical interpolation       (c) Distribution matched

Figure 5: 4-point interpolation between 4 sampled points (corners) from DCGAN trained on LSUN ($128 \times 128$) using a uniform prior. The same interpolation is shown using linear, SLERP and distribution matched interpolation.

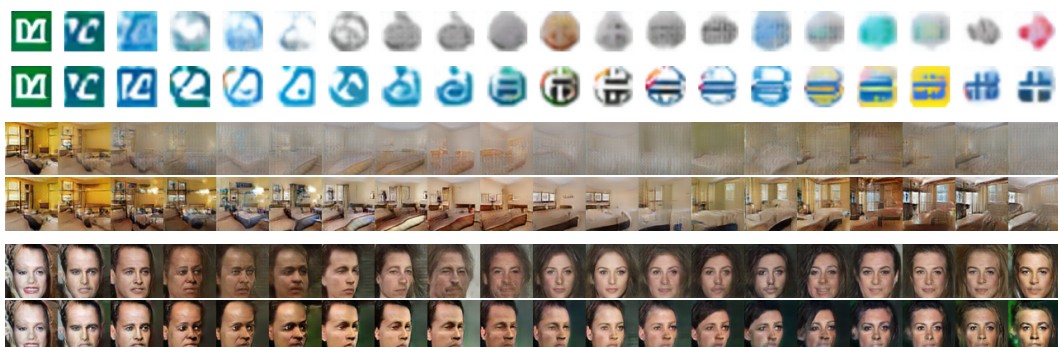

Figure 6: Random walk for LLD, LSUN (64 x 64) and CelebA. The random walks consist of a succession of steps in random directions, calculated for the same sequence of directions using (non-matched) vicinity sampling in the upper rows and our proposed matched vicinity sampling in the lower rows.

interpolation (our framework generalizes to more operations, see below and Appendix), the SLERP heuristic tends to work well in practice but we provide a principled alternative.

**4-point interpolation**    An even stronger effect can be observed when we do 4-point interpolation, showcased in Figure 5 (LSUN) and Figure 8 (LLD icons). The higher resolution of the LSUN output highlights the very apparent loss of detail and increasing prevalence of artifacts towards the midpoint in the linear version, compared to SLERP and our matched interpolation.

**Midpoints (Appendix)**    In all cases, the point where the interpolation methods diverge the most, is at the midpoint of the interpolation where $t = 0.5$. Thus we provide 25 such interpolation midpoints in Figures 11 (LLD icons) and 12 (LSUN) in the Appendix for direct comparison.

**Vicinity sampling (Appendix)**    Furthermore we provide two examples for vicinity sampling in Figures 9 and 10 in the Appendix. Analogous to the previous observations, the output under a linear operator lacks definition, sharpness and saturation when compared to both spherical and matched operators.

**Random walk**    An interesting property of our matched vicinity sampling is that we can obtain a *random walk* in the latent space by applying it repeatedly: we start at a point $y_0 = z$ drawn from the prior, and then obtain point $y_i$ by sampling a single point in the vicinity of $y_{i-1}$, using some fixed 'step size' $\epsilon$. We show an example of such a walk in Figure 6, using $\epsilon = 0.5$. As a result of the repeated application of the vicinity sampling operation, the divergence from the prior distribution in the non-matched case becomes stronger with each step, resulting in completely unrecognizable output images on the LSUN and LLD icon models.

## 5    CONCLUSIONS

We proposed a framework that fully eliminates the distribution mismatch in the common latent space operations used for generative models. Our approach uses optimal transport to minimally modify (in $l_1$ distance) the operations such that they fully preserve the prior distribution. We give analytical formulas of the resulting (matched) operations for various examples, which are easily implemented. The matched operators give a significantly higher quality samples compared to the originals, having the potential to become standard tools for evaluating and exploring generative models.

## ACKNOWLEDGMENTS

This work was partly supported by ETH Zurich General Fund (OK) and Nvidia through a hardware grant.

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

## A  ON THE CURSE OF DIMENSIONALITY AND GEOMETRIC OUTLIERS

We note that the analysis here can bee seen as a more rigorous version of an observation made by White (2016), who experimentally show that there is a significant difference between the average norm of the midpoint of linear interpolation and the points of the prior, for uniform and Gaussian distributions.

Suppose our latent space has a prior with $\boldsymbol{z} = [Z_1, \cdots, Z_d] \in [-1, 1]^d$ with i.i.d entries $Z_i \sim Z$. In this case, we can look at the squared norm

$$\|\boldsymbol{z}\|^2 = \sum_{i=1}^d Z_i^2. \tag{7}$$

From the Central Limit Theorem (CLT), we know that as $d \to \infty$,

$$\sqrt{d}(\frac{1}{d}\|\boldsymbol{z}\|^2 - \mu_{Z^2}) \to \mathcal{N}(0, \sigma_{Z^2}^2), \tag{8}$$

in distribution. Thus, assuming $d$ is large enough such that we are close to convergence, we can approximate the distribution of $\|\boldsymbol{z}\|^2$ as $\mathcal{N}(d\mu_{Z^2}, d\sigma_{Z^2}^2)$. In particular, this implies that almost all points lie on a relatively thin spherical shell, since the mean grows as $O(d)$ whereas the standard deviation grows only as $O(\sqrt{d})$.

We note that this property is well known for i.i.d Gaussian entries (see e.g. Ex. 6.14 in MacKay (2003)). For Uniform distribution on the hypercube it is also well known that the mass is concentrated in the corner points (which is consistent with the claim here since the corner points lie on a sphere).

Now consider an operator such as the midpoint of linear interpolation, $\boldsymbol{y} = \frac{1}{2}\boldsymbol{z}_1 + \frac{1}{2}\boldsymbol{z}_2$, with components $Y^{(i)} = \frac{1}{2}Z_1^{(i)} + \frac{1}{2}Z_2^{(i)}$. Furthermore, let's assume the component distribution $p_Z$ is symmetric around 0, such that $E[Z] = 0$.

In this case, we can compute:

$$E[(Y^{(i)})^2] = \text{Var}[\frac{1}{2}Z_1^{(i)} + \frac{1}{2}Z_2^{(i)}] = \frac{1}{2}\text{Var}[Z] = \frac{1}{2}\mu_{Z^2}^2 \tag{9}$$

$$\text{Var}[(Y^{(i)})^2] = \text{Var}[(\frac{1}{2}Z_1^{(i)} + \frac{1}{2}Z_2^{(i)})^2] = \frac{1}{4}\text{Var}[Z^2] = \frac{1}{4}\sigma_{Z^2}^2. \tag{10}$$

Thus, the distribution of $\|\boldsymbol{y}\|^2$ can be approximated with $\mathcal{N}(\frac{1}{2}d\mu_{Z^2}, \frac{1}{4}d\sigma_{Z^2}^2)$.

Therefore, $\boldsymbol{y}$ also mostly lies on a spherical shell, but with a different radius than $\boldsymbol{z}$. In fact, the shells will intersect at regions which have a vanishing probability for large $d$. In other words, when looking at the squared norm $\|\boldsymbol{y}\|^2$, $\|\boldsymbol{y}\|^2$ is a (strong) outlier with respect to the distribution of $\|\boldsymbol{z}\|^2$.

This can be quantified in terms of KL-Divergence:

$$D_{KL}(\|\boldsymbol{z}\|^2, \|\boldsymbol{y}\|^2) \approx D_{KL}(\mathcal{N}(d\mu_{Z^2}, d\sigma_{Z^2}^2), \mathcal{N}(\frac{1}{2}d\mu_{Z^2}, \frac{1}{4}d\sigma_{Z^2}^2)) \tag{11}$$

$$= \log \frac{\sqrt{d}\sigma_{Z^2}/2}{\sqrt{d}\sigma_{Z^2}} + \frac{d\sigma_{Z^2}^2 + (d\mu_{Z^2} - \frac{1}{2}d\mu_{Z^2})^2}{2\frac{1}{4}d\sigma_{Z^2}^2} - 1/2 \tag{12}$$

$$= d\frac{\mu_{Z^2}^2}{2\sigma_{Z^2}^2} - 1/2 - \log 2, \tag{13}$$

so $D_{KL}(\|\boldsymbol{z}\|^2, \|\boldsymbol{y}\|^2)$ grows linearly with the dimensions $d$.

### A.1  PROOF OF THEOREM 1

*Proof.* We will show it for the Kantorovich problem, the Monge version is similar.

Starting from (KP), we compute

$$inf\left\{E_{(\boldsymbol{x},\boldsymbol{y})\sim p_{\boldsymbol{x},\boldsymbol{y}}}[c(\boldsymbol{x},\boldsymbol{y})]\Big|(\boldsymbol{x},\boldsymbol{y})\sim p_{\boldsymbol{x},\boldsymbol{y}}, \boldsymbol{x}\sim p_{\boldsymbol{x}}, \boldsymbol{y}\sim p_{\boldsymbol{y}}\right\} \tag{14}$$

$$= inf\left\{E_{(\boldsymbol{x},\boldsymbol{y})\sim p_{\boldsymbol{x},\boldsymbol{y}}}[\sum_{i=1}^{d} C(\boldsymbol{x}^{(i)},\boldsymbol{y}^{(i)})]\Big|(\boldsymbol{x},\boldsymbol{y})\sim p_{\boldsymbol{x},\boldsymbol{y}}, \boldsymbol{x}\sim p_{\boldsymbol{x}}, \boldsymbol{y}\sim p_{\boldsymbol{y}}\right\} \tag{15}$$

$$= inf\left\{\sum_{i=1}^{d} E_{(\boldsymbol{x},\boldsymbol{y})\sim p_{\boldsymbol{x},\boldsymbol{y}}}[C(\boldsymbol{x}^{(i)},\boldsymbol{y}^{(i)})]\Big|(\boldsymbol{x},\boldsymbol{y})\sim p_{\boldsymbol{x},\boldsymbol{y}}, \boldsymbol{x}\sim p_{\boldsymbol{x}}, \boldsymbol{y}\sim p_{\boldsymbol{y}}\right\} \tag{16}$$

$$\geq \sum_{i=1}^{d} inf\left\{E_{(\boldsymbol{x},\boldsymbol{y})\sim p_{\boldsymbol{x},\boldsymbol{y}}}[C(\boldsymbol{x}^{(i)},\boldsymbol{y}^{(i)})]\Big|(\boldsymbol{x},\boldsymbol{y})\sim p_{\boldsymbol{x},\boldsymbol{y}}, \boldsymbol{x}\sim p_{\boldsymbol{x}}, \boldsymbol{y}\sim p_{\boldsymbol{y}}\right\} \tag{17}$$

$$= \sum_{i=1}^{d} inf\left\{E_{(X,Y)\sim p_{X,Y}}[C(X,Y)]\Big|(X,Y)\sim p_{X,Y}, X\sim p_X, Y\sim p_Y\right\} \tag{18}$$

$$= d\cdot inf\left\{E_{(X,Y)\sim p_{X,Y}}[C(X,Y)]\Big|(X,Y)\sim p_{X,Y}, X\sim p_X, Y\sim p_Y\right\}, \tag{19}$$

$$\tag{20}$$

where the inequality in (17) is due to each term being minimized separately.

Now let $\mathcal{P}_d(X,Y)$ be the set of joints $p_{\boldsymbol{x},\boldsymbol{y}}$ with $p_{\boldsymbol{x},\boldsymbol{y}}(x,y)=\prod_{i=1}^{d} p_{X,Y}(x^{(i)},y^{(i)})$ where $p_{X,Y}$ has marginals $p_X$ and $p_Y$. In this case $\mathcal{P}_d(X,Y)$ is a subset of all joints $p_{\boldsymbol{x},\boldsymbol{y}}$ with marginals $p_{\boldsymbol{x}}$ and $p_{\boldsymbol{y}}$, where the pairs $(\boldsymbol{x}^{(1)},\boldsymbol{y}^{(1)}),\ldots,(\boldsymbol{x}^{(d)},\boldsymbol{y}^{(d)}))$ are constrained to be i.i.d. Starting again from (16) can compute:

$$inf\left\{\sum_{i=1}^{d} E_{(\boldsymbol{x},\boldsymbol{y})\sim p_{\boldsymbol{x},\boldsymbol{y}}}[C(\boldsymbol{x}^{(i)},\boldsymbol{y}^{(i)})]\Big|(\boldsymbol{x},\boldsymbol{y})\sim p_{\boldsymbol{x},\boldsymbol{y}}, \boldsymbol{x}\sim p_{\boldsymbol{x}}, \boldsymbol{y}\sim p_{\boldsymbol{y}}\right\}$$

$$\leq inf\left\{\sum_{i=1}^{d} E_{(\boldsymbol{x},\boldsymbol{y})\sim p_{\boldsymbol{x},\boldsymbol{y}}}[C(\boldsymbol{x}^{(i)},\boldsymbol{y}^{(i)})]\Big|p_{\boldsymbol{x},\boldsymbol{y}}\in\mathcal{P}_d(X,Y)\right\} \tag{21}$$

$$= inf\left\{\sum_{i=1}^{d} E_{(\boldsymbol{x},\boldsymbol{y})\sim p_{\boldsymbol{x},\boldsymbol{y}}}[C(\boldsymbol{x}^{(i)},\boldsymbol{y}^{(i)})]\Big|p_{\boldsymbol{x},\boldsymbol{y}}\in\mathcal{P}_d(X,Y)\right\} \tag{22}$$

$$= inf\left\{\sum_{i=1}^{d} E_{(X,Y)\sim p_{X,Y}}[C(X,Y)]\Big|(X,Y)\sim p_{X,Y}, X\sim p_X, Y\sim p_Y\right\} \tag{23}$$

$$= d\cdot inf\left\{E_{(X,Y)\sim p_{X,Y}}[C(X,Y)]\Big|(X,Y)\sim p_{X,Y}, X\sim p_X, Y\sim p_Y\right\}, \tag{24}$$

$$\tag{25}$$

where the inequality in (21) is due to minimizing over a smaller set.

Since the two inequalities above are in the opposite direction, equality must hold for all of the expressions above, in particular:

$$inf\left\{E_{(\boldsymbol{x},\boldsymbol{y})\sim p_{\boldsymbol{x},\boldsymbol{y}}}[c(\boldsymbol{x},\boldsymbol{y})]\Big|(\boldsymbol{x},\boldsymbol{y})\sim p_{\boldsymbol{x},\boldsymbol{y}}, \boldsymbol{x}\sim p_{\boldsymbol{x}}, \boldsymbol{y}\sim p_{\boldsymbol{y}}\right\} \tag{26}$$

$$= d\cdot inf\left\{E_{(X,Y)\sim p_{X,Y}}[C(X,Y)]\Big|(X,Y)\sim p_{X,Y}, X\sim p_X, Y\sim p_Y\right\} \tag{27}$$

Thus, (KP) and (KP-1-D) equal up to a constant, and minimizing one will minimize the other. Therefore the minimization of the former can be done over $p_{X,Y}$ with $p_{\boldsymbol{x},\boldsymbol{y}}(x,y) = \prod_{i=1}^{d} p_{X,Y}(x^{(i)},y^{(i)})$. □

# B    CALCULATIONS FOR EXAMPLES

In the next sections, we illustrate how to compute the matched operations for a few examples, in particular for linear interpolation and vicinity sampling, using a uniform or a Gaussian prior. We picked the examples where we can analytically compute the uniform transport map, but note that it is also easy to compute $F_Z^{[-1]}$ and $(F_Y(y))$ numerically, since one only needs to estimate CDFs in one dimension.

Since the components of all random variables in these examples are i.i.d, for such a random vector $\boldsymbol{x}$ we will implicitly write $X$ for a scalar random variable that has the distribution of the components of $\boldsymbol{x}$.

When computing the monotone transport map $T_{X \to Y}^{\text{mon}}$, the following Lemma is helpful.

**Lemma 1** (Theorem 2.5 in Santambrogio (2015)). *Suppose a mapping $g(x)$ is non-decreasing and maps a continuous distribution $p_X$ to a distribution $p_Y$, i.e.*

$$g(X) \sim Y, \tag{28}$$

*then $g$ is the monotone transport map $T_{X \to Y}^{mon}$.*

According to Lemma 1, an alternative way of computing $T_{X \to Y}^{\text{mon}}$ is to find some $g$ that is non-decreasing and transforms $p_X$ to $p_Y$.

EXAMPLE 1:UNIFORM LINEAR INTERPOLATION

Suppose $\boldsymbol{z}$ has uniform components $Z \sim \text{Uniform}(-1, 1)$. In this case, $p_Z(z) = 1/2$ for $-1 < z < 1$.

Now let $\boldsymbol{y}_t = t\boldsymbol{z}_1 + (1 - t)\boldsymbol{z}_2$ denote the linear interpolation between two points $\boldsymbol{z}_1, \boldsymbol{z}_2$, with component distribution $p_{Y_t}$. Due to symmetry we can assume that $t > 1/2$, since $p_{Y_t} = p_{Y_{1-t}}$. We then obtain $p_{Y_t}$ as the convolution of $p_{tZ}$ and $p_{(1-t)Z}$, i.e. $p_{Y_t} = p_{tZ} * p_{(1-t)Z}$. First we note that $p_{tZ} = 1/(2t)$ for $-t < z < t$ and $p_{(1-t)Z} = 1/(2(1-t))$ for $-(1-t) < z < 1-t$. We can then compute:

$$p_{Y_t}(y) = (p_{tZ} * p_{(1-t)Z})(y) \tag{29}$$

$$= \frac{1}{2(1-t)(2t)} \begin{cases} 0 & \text{if } y < -1 \\ y + 1 & \text{if } -1 < y < -t + (1-t) \\ 2 - 2t & \text{if } -t + (1-t) < y < t - (1-t) \\ -y + 1 & \text{if } t - (1-t) < y < 1 \\ 0 & \text{if } 1 < y \end{cases} \tag{30}$$

$$\tag{31}$$

The CDF $F_{Y_t}$ is then obtained by computing

$$F_{Y_t}(y) = \int_{-\infty}^{y} p_{Y_t}(y')dy' \tag{32}$$

$$= \frac{1}{2(1-t)(2t)} \begin{cases} 0 & \text{if } y < -1 \\ \frac{1}{2}(y + 1)(y + 1) & \text{if } -1 < y < 1 - 2t \\ 2(1-t)(y + t) & \text{if } 1 - 2t < y < 2t - 1 \\ 2(1-t)(3t - 1) + (-\frac{1}{2}y^2 + y + \frac{1}{2}(2t - 1)^2 - (2t - 1)) & \text{if } 2t - 1 < y < 1 \\ 2(1-t)(2t) & \text{if } 1 < y \end{cases} \tag{33}$$

Since $p_Z(z) = 1/2$ for $|z| < 1$, we have $F_Z(z) = \frac{1}{2}z + \frac{1}{2}$ for $|z| < 1$. This gives $F_Z^{[-1]}(p) = 2(p - \frac{1}{2})$.

Now, we just compose the two mappings to obtain $T_{Y_t \to Z}^{\text{mon}}(y) = F_Z^{[-1]}(F_{Y_t}(y))$.

EXAMPLE 2: UNIFORM VICINITY SAMPLING AND RANDOM WALK

Let $z$ again have uniform components on $[-1, 1]$. For vicinity sampling, we want to obtain new points $z'_1, \cdot, z'_k$ which are close to $z$. We thus define

$$z'_i := z + \epsilon u_i, \tag{34}$$

where $u_i$ also has uniform components, such that each coordinate of $z'_i$ differs at most by $\epsilon$ from $z$. By identifying $tZ'_i = tZ + (1-t)U_i$ with $t = 1/(1+\epsilon)$, we see that $tZ'_i$ has identical distribution to the linear interpolation $Y_t$ in the previous example. Thus $g_t(Z'_i) := T^{\mathrm{mon}}_{Y_t \to Z}(tZ'_i)$ will have the distribution of $Z$, and by Lemma1 is then the monotone transport map from $Z'_i$ to $Z$.

EXAMPLE 3: GAUSSIAN LINEAR INTERPOLATION, VICINITY SAMPLING AND ANALOGIES

Suppose $z$ has components $Z \sim \mathcal{N}(0, \sigma^2)$. In this case, we can compute linear interpolation as before, $y_t = tz_1 + (1-t)z_2$. Since the sum of Gaussians is Gaussian, we get, $Y_t \sim \mathcal{N}(0, t^2\sigma^2 + (1-t)^2\sigma^2)$. Now, it is easy to see that with a proper scaling factor, we can adjust the variance of $Y_t$ back to $\sigma^2$. That is, $\frac{1}{\sqrt{t^2+(1-t)^2}}Y_t \sim \mathcal{N}(0, \sigma^2)$, so by Lemma 1 $g_t(y) := \frac{1}{\sqrt{t^2+(1-t)^2}}y$ is the monotone transport map from $Y_t$ to $Z$.

By adjusting the vicinity sampling operation to

$$z'_i := z + \epsilon e_i, \tag{35}$$

where $e_i \sim \mathcal{N}(0, 1)$, we can similarly find the monotone transport map $g_\epsilon(y) = \frac{1}{\sqrt{1+\epsilon^2}}y$.

Another operation which has been used in the literature is the "analogy", where from samples $z_1, z_2, z_3$, one wants to apply the difference between $z_1$ and $z_2$, to $z_3$. The transport map is then $g(y) = \frac{1}{\sqrt{3}}y$

## C  DETERMINISTIC VS STOCHASTIC OPERATIONS

In Strategy 1, we only considered deterministic mappings $f$ such that $\tilde{y} = f(y)$ recovers the prior distribution. However, one can also consider stochastic mappings $f$. One example for linear interpolation, proposed by the area chair (AC) [1], is to set $\tilde{y}|z_1, z_2 \sim \mathcal{N}(tz_1 + (1-t)z_2, (1 - t^2 - (1-t)^2)\sigma^2)$ for the case when $z_1, z_2 \sim \mathcal{N}(\mu, \sigma^2)$ are Gaussian. This ensures that marginally $\tilde{y} \sim \mathcal{N}(\mu, \sigma^2)$, making it a valid (stochastic) modification to $y$ that recovers the prior distribution.

However, our matched interpolation has two benefits over this approach: since it is deterministic (and continuous), it means that the line between $z_1, z_2$ gets mapped to a smooth trajectory in the latent space. Furthermore, for the cost $c(x, y) := \|x - y\|_1$, it is optimal (see Theorem 2 ), even for the (KP) problem where $f$ is allowed to be stochastic.

In Figure 7 we illustrate the differences between the two approaches, using a WGAN-GP model trained on CelebA with a 128 dimensional $\mathcal{N}(0, 1)$ latent space. We see that like our matched interpolation, the stochastic one suggested by the AC has better samples than the linear interpolation. However, due to the stochastic nature, adjacent data points do not change smoothly. In contrast our matched interpolation warps the trajectory of the linear one, which ensures that the resulting trajectory remains smooth.

---

[1] https://openreview.net/forum?id=BklCusRct7&noteId=Hkg6swibeN

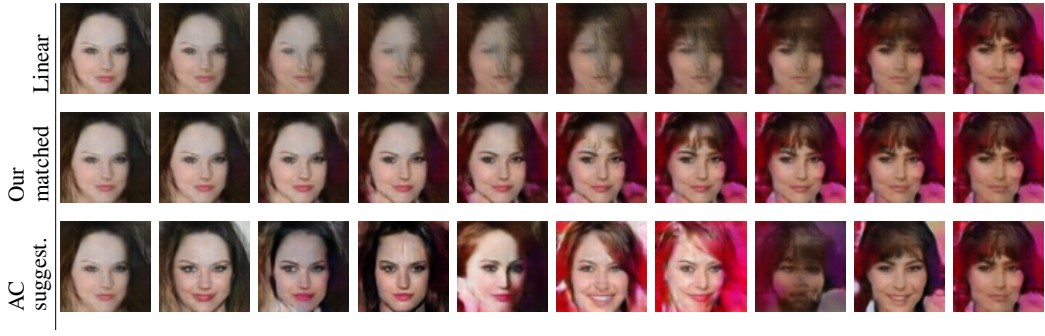

Figure 7: We illustrate the difference between linear interpolation, our matched interpolation and the stochastic interpolation suggested by the area chair.

# D ADDITIONAL EXPERIMENTS

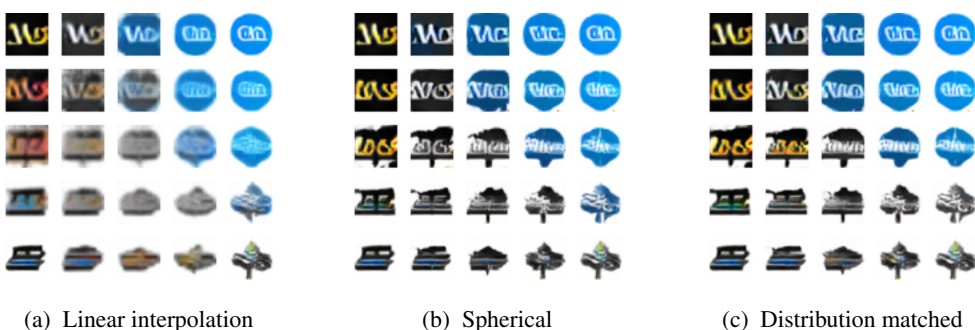

(a) Linear interpolation      (b) Spherical      (c) Distribution matched

Figure 8: 4-point interpolation between 4 sampled points (corners) from DCGAN trained on icon dataset using a uniform prior. The same interpolation is shown using linear, SLERP and distribution matched interpolation.

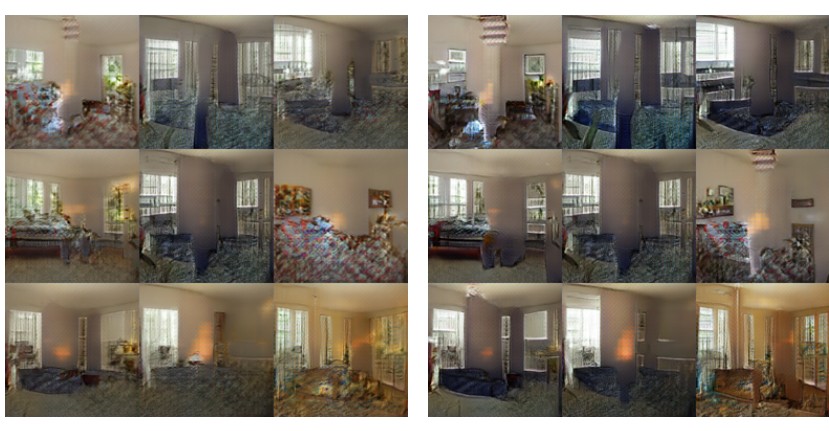

(a) Vicinity sampling      (b) Matched vicinity sampling

Figure 9: Vicinity sampling on LSUN dataset ($128 \times 128$) with uniform prior. The sample in the middle is perturbed in random directions producing the surrounding sample points.

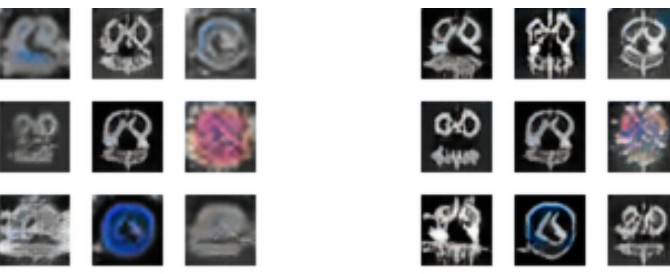

(a) Vicinity sampling      (b) Matched vicinity sampling

Figure 10: Vicinity sampling on LLD icon dataset with uniform prior. The sample in the middle is perturbed in random directions producing the surrounding sample points.

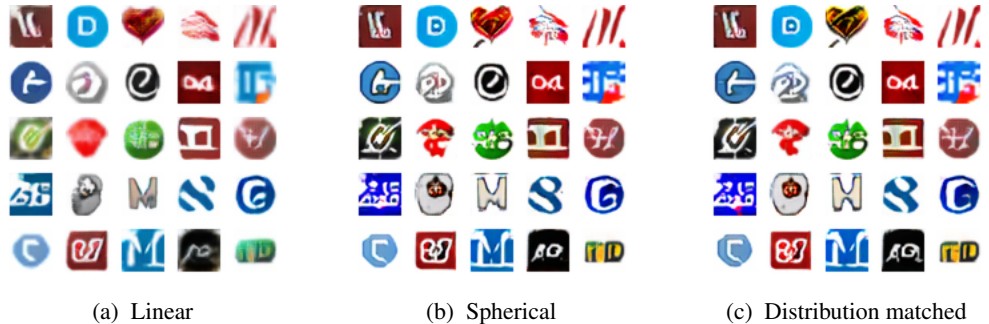

(a) Linear      (b) Spherical      (c) Distribution matched

Figure 11: Midpoint sampling for linear, SLERP and uniform-matched interpolation when using the same pairs of sample points on LLD icon dataset with uniform prior.

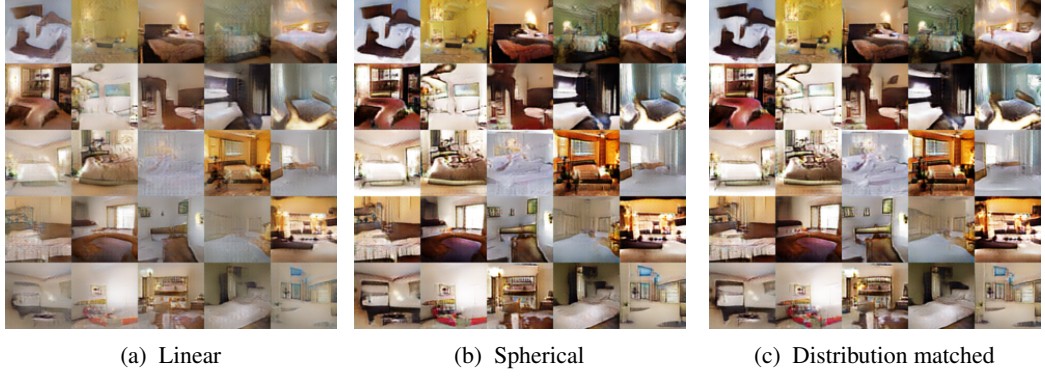

(a) Linear      (b) Spherical      (c) Distribution matched

Figure 12: Midpoint sampling for linear, SLERP and uniform-matched interpolation when using the same pairs of sample points on LSUN ($64 \times 64$) with uniform prior.

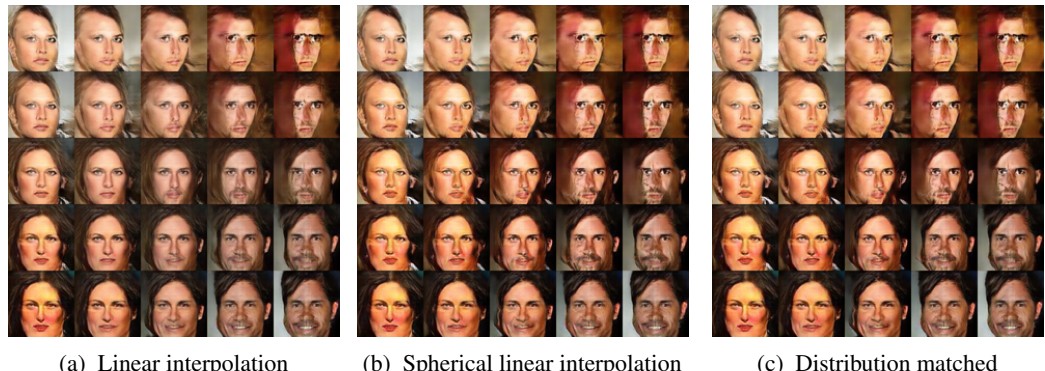

(a) Linear interpolation    (b) Spherical linear interpolation    (c) Distribution matched

Figure 13: 4-point interpolation between 4 sampled points (corners) from DCGAN trained on CelebA with Gaussian prior. The same interpolation is shown using linear, SLERP and distribution matched interpolation.

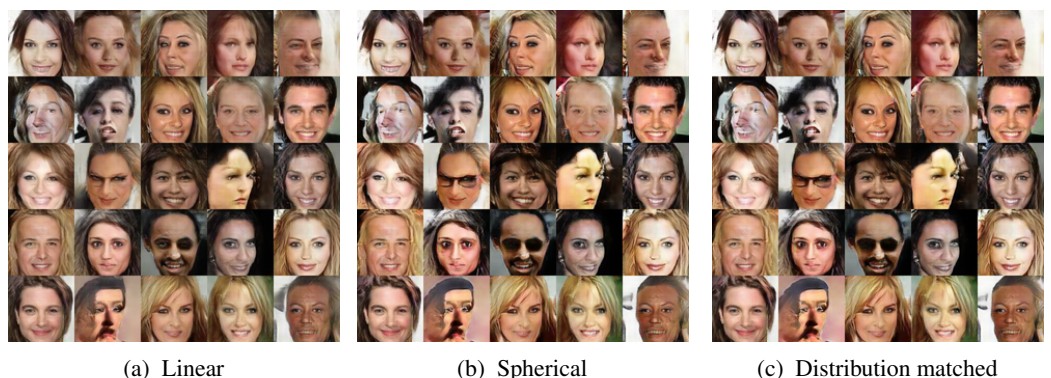

(a) Linear    (b) Spherical    (c) Distribution matched

Figure 14: Midpoint sampling for linear, SLERP and uniform-matched interpolation when using the same pairs of sample points on CelebA with Gaussian prior.

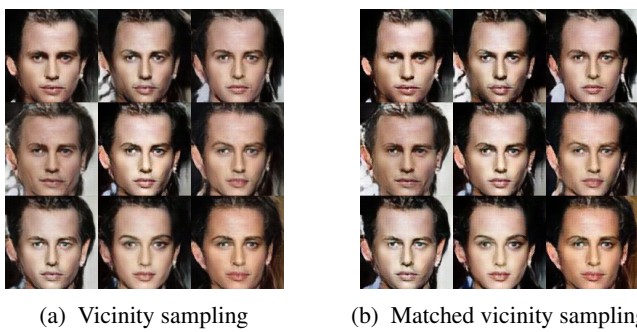

(a) Vicinity sampling    (b) Matched vicinity sampling

Figure 15: Vicinity sampling on CelebA dataset with Gaussian prior. The sample in the middle is perturbed in random directions producing the surrounding sample points.

