# OpenReview forum: "Optimal Transport Maps For Distribution Preserving Operations on Latent Spaces of Generative Models"
_ICLR.cc/2019/Conference_

### Official Review · AnonReviewer1 · 2018-10-31
**A clear explanation on distribution mismatch in generative models but needs stronger motivation**

**Rating:** 5
**Confidence:** 3

**Review:**

The paper addresses the latent space distribution mismatch in VAEs and GANs. The authors try to solve the issue by optimal transport theory and the proposed method on the latent space yields better quality in the generated samples.

To me, the motivation is not very strong. In DCGAN, amazingly, latent space linear operations can carry over to the generated images. But it’s not something people are usually concerned with in GANs.  I understand that latent space operations can provide insights into how the trained generator works. But how can it improve the actual GAN training? Choosing Gaussian or uniform distribution for the latent variable is mainly for ease of computation and I am not sure if the motivation to match the distributions is very strong in GAN applications. Perhaps it more important in the context of VAEs.

At the first glance, the proposed form of transformation is not surprising. Though optimal transport is a very powerful theoretical tool, it serves more like an explanation or validation, rather than the motivation. I felt the theory part could be simper.

In the quantitative comparisons with other methods, all simulations seem to be in the context of GAN. The difference in 2-point cases (table 2) is not significant and the author only compares with linear interpolation but not SLERP. I would like to see more quantitative comparisons with other methods and also some empirical studies in the context of VAEs.

---

> ### Author Response · Authors · 2018-11-27
> **Response**
>
> We thank the reviewer for the feedback.
>
> We argue that just because latent space operations do not help with GAN training, it does not mean they are not useful. Just as the reviewer suggests, they provide insights into how the trained generator works. For example, interpolations are one of the most intuitive ways to illustrate whether a model is capable of synthesizing new images (instead of just memorizing) and to visualize the learned latent space.
> Many of the most impactful papers on generative models have employed such methods, such as:
>
> Auto-Encoding Variational Bayes (Kingma and Welling, 2013)
> Generative Adversarial Networks (Goodfellow et al. 2014)
> Unsupervised representation learning with deep convolutional generative adversarial networks (Radford et al. 2015)
> Progressive Growing of GANs for Improved Quality, Stability, and Variation (Karras et al. 2018) ( https://www.youtube.com/watch?v=G06dEcZ-QTg&feature=youtu.be&t=77 )
> Large Scale Gan Training For High Fidelity Natural Image Synthesis (Brock et al, 2018)
>
> Our framework is agnostic to the choice of latent space (as long at is has i.i.d. components), if you change the latent space our "matched" operations just need to be recomputed. This choice might often be based on convenience (or convention), but for VAEs it is also important that the KL-divergence is readily computed between the encoded distribution and the latent prior (which typically motivates the choice of a gaussian latent space).
>
> We are confused by the claim that our proposed approach is "unsurprising" and that "it serves more like an explanation or validation, rather than the motivation". Could the reviewer elaborate on this? No prior works approached this problem from this direction and our proposed approach is not at all obvious.
>
> The point of Table 2. is not to serve as some kind of "benchmark" for interpolation methods, but to illustrate that the distribution mismatch of the original operations results in significant drops in Inception Scores, and that our method fully eliminates this drop (obtaining the same scores as when generating random samples). We are confused by the claim that the the differences are "not significant" given that the drop for the original linear operations is up to 29% whereas we fully recover the performance of the original model (when sampled randomly).
> We discussed and compared visually with SLERP (a heuristic designed for 2-point interpolation) in Sec. 4.2, and explicitly stated that we do not provide a "better" method in terms of visual quality, but stress that the goal of this work is to construct operations in a principled manner, whose samples are consistent with the generative model.
> We computed the Inception Scores for SLERP for 2-point interpolation, and the results are consistent with this, where it gives similar scores as our proposed approach: 7.89 +- 0.09, 3.68 +- 0.09, 3.90 +- 0.11 and 2.04 +- 0.04 for CIFAR-10, LLD-icon, LSUN and CelebA respectively. Furthermore, we note that 2-point interpolation is just one of the many operations our framework can be applied to.

---

### Official Review · AnonReviewer2 · 2018-11-05
**Interesting paper on distribution matching between training & test data in GAN; more evidence needed**

**Rating:** 5
**Confidence:** 3

**Review:**

This paper considers the issue of distribution mismatch between the input data used for training generative models and the new data for new instance generation. Given a sample operation, the authors propose to use the so-called optimal transport to map the distribution of the new data to that of the input data that were used training. The optimal transport is essentially a monotonic transformation as the composite of the inverse of the target distribution and the source distribution.

The paper is in general well written. However, I am concerned with two issues here, which are related to the motivation and performance evaluation, respectively. First, the authors didn't make it clear what data generation of the trained generative model suffers from the distribution mismatch issue, although there was some discussion on this in the literature, as the authors mentioned. To me, once the generative model is successfully trained, it is something like a physical process, and new data, which are contained in the support of the training data, can always be used as input to generate new data. (Personally, I think this is very different from covariate shift correction in domain adaptation, in which the correction is necessary because simpler models, instead of flexible, nonparametric ones, are used to make prediction.) Second, the authors used the Inception Score for performance evaluation. Please give this score in the paper and make its definition clear. To me, it is not surprising at all that the proposed method had a better Inception Score: roughly speaking, when we use interpolated values of the training input data to generate images, the Inception Score is expected to decrease, compared to that evaluated on the training data. Intuitively, a very high Inception Score may indicate that we are not trying to generalize, but just memorize the training input data. An explanation about this point would be highly appreciated.

---

> ### Author Response · Authors · 2018-11-27
> **Response**
>
> We thank the reviewer for the feedback.
>
> Regarding the data-generation process: we do use a model that is only trained once to generate new data.
> However, we observe (both theoretically and experimentally) the opposite of what you claim: even though you train on a specific distribution (say uniform in the 100 dimensional hypercube), it matters where from the support you sample. Of course, if you use the model as a "physical process" and sample new data with the same distribution as you used during training, you do not have a problem. However, once you start sampling the distribution in a different way (e.g. by interpolating between samples), even though you remain in the support of your distribution you start getting "abnormal" latent codes which your model performs poorly on. We urge the reviewer to carefully look at Figure 2. in the paper, which illustrates how different (geometrically) the interpolated samples can be compared to the endpoints, due to the high dimensionality of the space.
>
> Regarding the Inception Score, it was proposed in by Salimans et al. ( https://arxiv.org/pdf/1606.03498.pdf ), and will describe it better in the paper.
> We do not understand your statement that "when we use interpolated values of the training input data to generate images, the Inception Score is expected to decrease, compared to that evaluated on the training data".
> We are not interpolating training input data, we are interpolating random latent points during evaluation, the exact same latent points that are used when evaluating the model in its standard setting. We do not obtain improved Inception Scores compared to the original model (when sampled randomly), rather we avoid dropping in performance as happens when you linearly interpolate.

---

### Official Review · AnonReviewer3 · 2018-11-05
**A neat and sound interpolation modification approach but more experiments are needed**

**Rating:** 7
**Confidence:** 5

**Review:**

Noticing that widely used latent code interpolations for exploring the generative capabilities of VAEs and GANs have distribution mismatch problems, this paper proposes to utilize monotone transport map to exactly eliminate the distribution mismatch between modified interpolated codes and a prior distribution, assuming I.I.D. code components and a L1 code distance. More precisely, a transformation of the latent space operation is learnt with the objective that the distribution of the transformed variable match the prior distribution used in training the generative models. Optimal transport is used as a measure to minimize the two distributions. By restricting the class of cost functions used in the optimal transport formulation, the solution to the optimal transport problem (and hence the transformation function) has been shown to take a simple form (closed form in cases where cdf has a analytical form). Experiments on CIFAR-10, LLD-icon, LSUN, CelebA datasets show that, the minimally modified interpolated codes for several different interpolations produce samples with higher Inception Scores and better visual effects under an improved Wasserstein GAN than the original interpolated codes.

This paper is well written, the studied problem is highly important, and the approach presented has potentially wide applications.

However, there are some concerns about the experimental evaluations,

1. Although the quantitative evaluations for 2-point and 4-point interpolations are important, it is hard to assess these interpolations in a semantically meaningful way. Extensive quantitative (FID and IS) and qualitative evaluations should be conducted for analogy interpolations. For example, adding glasses, adding mustache, and many others. It is much easier to assess the quality of the generated images from the minimally modified interpolated code for this category in a meaningful way.

2. Another concern is that how big the effect of the transformation function inducing on the latent space operations will be. For example, a linear interpolation is no longer linear after getting transformed. So, are there transformations that drastically transform the original latent space operations? In that case, will the transformed variable make any sense with respect to the original latent space operations? Extensive experiments for analogy interpolations are required to answer these questions.

3. Experiments have been shown only on GAN architectures, however, the framework can be easily extended to VAEs. Experiments on VAEs will be informative.

Minor:

Section 1.1, in the second paragraph, (SLERP) should be moved a correct position.

Figure 2: it's better to use a different color for midpoint linear other than blue

Problem 1, f* ---> f*:

---

> ### Author Response · Authors · 2018-11-27
> **Response**
>
> We thank the reviewer for the feedback!
>
> Regarding the evaluation of analogy interpolations, we did not due this due to the the added complexity involved. In particular, we there is no standardized way of performing analogies in terms of how to select the examples and the difference vectors. It can be done over averages of groups of samples or over individual samples. In both cases, how the samples are produced (e.g. manually selecting them or using a conditional GAN) would also need to be taken into account. Nonetheless, we think it would be an interesting in the future to explore the application our framework for analogies - and see nothing that prevents its use in principle.
>
> Regarding the effect of the transformation on the operation:  we agree that this is a valid concern. In our framework, we take the perspective that the desired output distribution is the same as the original latent distribution, and search for the minimal perturbation (in l1 distance). While a natural approach, the required (minimal) perturbation could still be large. To assess this, we intend to add to the paper the average effect of the perturbation for the experiments in Table 2 (i.e. the l1 distance between the original and modified code samples). This way, we can quantify how much the operations are impacted by the adjustment.
>
> We agree that experiments on VAEs would well complement the paper, but we expect the results to be the same. In particular, since our approach fully eliminates the distribution mismatch, we are guaranteed to get the same sample quality as from random samples. The only question remaining (which is still interesting) is whether VAEs are more or less sensitive to the mismatch when using the unmodified operations.

---

### Meta-Review · Area_Chair1 · 2018-12-14
**Smooth latent space interpolation for deep generative models**

**Confidence:** 4
**Recommendation:** Accept (Poster)

**Metareview:**

This is a well-written paper that shows how to use optimal transport to perform smooth interpolation, between two random vectors sampled from the prior distribution of the latent space of a deep generative model. By encouraging the marginal of the interpolated vector to match the prior distribution, these interpolated distribution-preserving random vectors in the latent space are shown to result in better image interpolation quality for GANs. The problem is of interest to the community and the resulted solutions are simple to implement.

As pointed out by Reviewer 1, the paper could be made clearly more convincing by showing that these distribution preservation operations also help perform interpolation in the latent space of VAEs, and the AC strongly encourages the authors to add these results if possible.

The AC appreciates that the authors have added experiments to satisfactorily address his/her concern:

"Suppose z_1,z_2 are independent, and drawn from N(\mu,\Sigma), then t z_1 + (1-t)z_2 ~ N(\mu, (t^2+(1-t)^2)\Sigma). If one lets y | z_1, z_2 ~ N(t z_1 + (1-t)z_2, (1-t^2-(1-t)^2)\Sigma) as the latent space interpolation, then marginally we have y ~ N(\mu, \Sigma). This is an extremely simple and fast procedure to make sure that the latent space interpolation y is highly related to the linear interpolation t z_1 + (1-t)z_2 but also satisfies  y ~ N(\mu, \Sigma)."

The AC strongly encourages the authors to add these new results into their revision, and highlight "smooth interpolation" as an important characteristic in addition to "distribution preserving." A potential suggestion is changing "Distribution Preserving Operations" in the title to "Distribution Preserving Smooth Operations."